# Potential of Fatty Acids in Treating Sarcopenia: A Systematic Review

**DOI:** 10.3390/nu15163613

**Published:** 2023-08-17

**Authors:** Tao Huang, Chaoran Liu, Can Cui, Ning Zhang, Wing Hoi Cheung, Ronald Man Yeung Wong

**Affiliations:** Department of Orthopaedics & Traumatology, The Chinese University of Hong Kong, Hong Kong SAR, China; sthtao0623@163.com (T.H.); liuchaoran927@link.cuhk.edu.hk (C.L.); cuican@link.cuhk.edu.hk (C.C.); zhangning@link.cuhk.edu.hk (N.Z.); louischeung@cuhk.edu.hk (W.H.C.)

**Keywords:** fatty acid, muscle, sarcopenia

## Abstract

This paper presents a systematic review of studies investigating the effects of fatty acid supplementation in potentially preventing and treating sarcopenia. PubMed, Embase, and Web of Science databases were searched using the keywords ‘fatty acid’ and ‘sarcopenia’. Results: A total of 14 clinical and 11 pre-clinical (including cell and animal studies) studies were included. Of the 14 clinical studies, 12 used omega-3 polyunsaturated fatty acids (PUFAs) as supplements, 1 study used ALA and 1 study used CLA. Seven studies combined the use of fatty acid with resistant exercises. Fatty acids were found to have a positive effect in eight studies and they had no significant outcome in six studies. The seven studies that incorporated exercise found that fatty acids had a better impact on elderlies. Four animal studies used novel fatty acids including eicosapentaenoic acid, trans-fatty acid, and olive leaf extraction as interventions. Three animal and four cell experiment studies revealed the possible mechanisms of how fatty acids affect muscles by improving regenerative capacity, reducing oxidative stress, mitochondrial and peroxisomal dysfunctions, and attenuating cell death. Conclusion: Fatty acids have proven their value in improving sarcopenia in pre-clinical experiments. However, current clinical studies show controversial results for its role on muscle, and thus the mechanisms need to be studied further. In the future, more well-designed randomized controlled trials are required to assess the effectiveness of using fatty acids in humans.

## 1. Introduction

Sarcopenia is an age-related degenerative disease that manifests itself with a loss of muscle mass, strength, and physical performance [1]. Patients with sarcopenia often suffer from frailty as well, leading to an increased risk of falls, fractures, and mortality [2]. Optimal care and treatment for sarcopenia are warranted to improve clinical outcomes and decrease socioeconomic burdens [3]. The prevalence of sarcopenia in adults aged 65 years or above has been shown to reach up to 40% based on the Asian Working Group for Sarcopenia (AWGS) 2019 guidelines [4]. A 2.3-fold increased risk of falls is reported for sarcopenic elderlies compared to the non-sarcopenic elders [5]. In patients with a fragility fracture [6,7], sarcopenia has been reported to reach up to 95% in males and 64% in females [8]. Recent studies have also shown that sarcopenia treatments in the United Kingdom (UK) costs approximately Great Britain Pound (GBP) 2.5 billion annually [2,9]. In the United States (US), the total annual estimated cost of hospitalization in people with sarcopenia is USD 40.4 billion, causing a substantial socioeconomic impact [10].

As of now, there is no U.S. Food and Drug Administration (FDA) approved drug to treat sarcopenia. The mainstay of sarcopenia treatment is physical exercise with resistance training and adequate nutrition [11,12]. However, older adults, particularly frail older adults, cannot well tolerate physical exercises in the long term [13]. Furthermore, besides physical training, clinical guidelines have recommended that elderlies have at least 1.0–1.2 g of protein/kilogram body weight to maintain skeletal muscle anabolism [14]. However, a high-protein diet may also aggravate existing metabolic diseases, including diabetes mellitus [15]. Recent research has proven that certain modifiable dietary factors for sarcopenia appear to be essential for its management. There is also increasing evidence that the use of fatty acids can positively affect muscle metabolism [16,17]. Recent studies have shown that fatty acids regulate skeletal muscle mass and muscle function. Chen et al. [18] found that long-chain fatty acids (LCFAs) were involved in the pathogenesis of sarcopenia by activating the apoptotic signaling, whilst polyunsaturated fatty acids (PUFAs) were related to the hypertrophy of muscle [19] and improved mitochondrial oxidative and antioxidant capacity [20,21]. A randomized controlled trial study showed that treatment with 4 g/day n3-PUFA could increase muscle strength by attenuating inflammation and may also improve exercise responsiveness in elderly females [22]. In addition to the effect on the inflammatory response, there is also increasing evidence that omega-3 fatty acids impact muscle protein synthesis by affecting the mammalian target of rapamycin (mTOR) signaling, which may increase muscle mass in older adults [11].

There is currently a lack of systematic reviews on the postulated mechanisms of fatty acid in people with sarcopenia [23,24,25]. This systematic review aimed to assess the effect of fatty acids in preventing and treating sarcopenia and to understand the underlying mechanisms.

## 2. Materials and Methods

### 2.1. Search Strategy 

This systematic review was performed according to the PRISMA guidelines [26]. The keywords for the search were “sarcopenia”, “fatty acids”, “lipid”, “essential fatty acid”, “hydroxy fatty acid”, “long chain fatty acid”, “medium chain fatty acid”, “short chain fatty acid”, “unsaturated fatty acid”, “treatment”, “therapy”, “diet therapy” and “drug therapy” in the electronic databases of PubMed, Web of Science, and Embase, which were searched until April 2023 (for full search strategy see Appendix A). 

### 2.2. Selection Criteria

Both clinical and pre-clinical studies were screened. The inclusion criteria for the clinical studies were (i) elderly patients aged 60 years old or above and diagnosed with sarcopenia, (ii) studies on fatty acid and its effect on sarcopenia, and (iii) randomized controlled trials. The exclusion criteria were (i) studies that did not research fatty acids or sarcopenia, (ii) studies not focusing on primary sarcopenia, (iii) review articles, (iv) conferences or abstract publications, and (v) non-English articles. 

The inclusion criteria for the pre-clinical studies were (i) studies on fatty acid and its effect on muscle cell lines or aged animals, and (ii) original research. The exclusion criteria were (i) studies not involving fatty acids or sarcopenia, (ii) conferences or abstract publications, (iii) review articles, and (iv) non-English articles. 

### 2.3. Study Selection

Titles and abstracts were screened following the search. A full-text screening was performed for the eligible articles based on the inclusion and exclusion criteria. Studies were screened independently by two reviewers (HT and LCR). Any disagreements that emerged were solved after a discussion with a third reviewer (WRMY). 

### 2.4. Risk of Bias Assessment

Risk of bias (RoB) assessment was implemented through the Cochrane RoB 2 tool for Randomized Controlled Trials (RCTs) following the Cochrane Handbook for Systematic Reviews of Interventions [27]. Two independent reviewers assessed each study for bias (refer to Appendix A).

### 2.5. Data Extraction and Analysis

Two reviewers (HT and LCR) performed data extraction, and disagreements were resolved after discussions with a third reviewer (WRMY). The data extracted from clinical studies were author, year of publication, subjects, intervention, primary outcomes (muscle mass and muscle function), secondary outcomes (physical performance), and key findings. For pre-clinical studies, the extracted data were animal models or cell lines, interventions, endpoints, primary outcomes (muscle mass and function), secondary outcomes (mechanisms and changes in cytokines), and key findings. Qualitative analysis was performed in this review due to the heterogeneity in both the clinical and pre-clinical studies. The findings were evaluated based on the results of the included studies. The Jadad scale [28] was used to evaluate the quality of randomized controlled trials. The scores on the Jadad scale were marked as 5 (high), 4 (good), 3 (medium), and 0–2 (low) quality. Only studies with a Jadad scale score of >3 (see Appendix A) were considered.

## 3. Results

The PRISMA flowchart is shown in Figure 1. A total of 505 articles were extracted (PubMed: 285, Embase: 19, and Web of Science: 201). After removing the duplicate records (*n* = 69), a total of 436 studies were identified for the title and abstract screening. Based on the inclusion and exclusion criteria, 316 studies were excluded, and 120 full-text articles were assessed for eligibility. Finally, 23 studies (13 clinical and 10 pre-clinical) were included for qualitative analysis (Figure 1).

### 3.1. Clinical Studies

Table 1 shows the summary of the 13 clinical studies. All studies were randomized controlled trials including 16 to 1680 participants [23,24,25,29,30,31,32,33,34,35,36,37,38] published between 2009 and 2023. Eleven studies used omega-3/n-3 PUFA [23,24,25,29,31,32,33,34,35,36,38], one study used Alpha-linolenic acids (ALAs) [30], and one study used conjugated linoleic acids (CLAs) [37] as supplements for the intervention. One study [23] used the diagnostic criteria of sarcopenia based on the European Working Group on Sarcopenia (EWGSOP) [39]. One study tested low muscle mass [34] based on older definitions that only used the assessment of muscle mass. The remaining studies recruited adults 60 years or older as participants. 

#### 3.1.1. Outcome Assessments

A total of 2128 elderly participants (60–85 years old, 59.39% male) were included. Assessment outcomes include muscle mass, muscle strength, physical performance, blood sampling, and muscle biopsy. One study did not test muscle mass or physical performance but only focused on muscle biopsy [24]. Nine studies not only tested muscle mass and physical performance but also collected blood samples to test for interleukin 6 (IL-6) and tumor necrosis factor-α (TNF-α) levels [24,30,31,32,33,35,37,38]. Two studies assessed muscle and myofibrillar protein synthesis rates to see the effect of fatty acids on power at the metabolic level [35,37]. 

Dual-energy X-ray absorptiometry (DXA) scan was used in six studies [23,30,31,35,36,37], whilst magnetic resonance imaging (MRI) was used in four studies [25,29,32,36]; some studies used bioimpedance impedance analysis (BIA) [34], and computed tomography (CT) scan [33] to evaluate muscle mass. Two studies did not perform physical performance measurement [24,35], and some studies measured muscle strength [23,25,30,32,33,34,35,36,37,38], habitual physical ability [23,25,29,32,33,36], and endurance [29,31,33,34,36,38] for assessing physical performance. These results were used as the criteria for evaluating intervention outcomes.

#### 3.1.2. Intervention and Results 

During the intervention with fatty acids, four studies [24,30,31,32] added extra resistance exercise training 2–3 times per week. Four studies [25,29,34,37] used other supplements, including vitamin D, vitamin E, and whey protein isolate capsules (WPIs), in the intervention group. Positive outcomes were shown in six studies [23,24,29,30,36,38] with improved muscle mass and physical performance using the fatty acids with resistance exercise or other nutritional supplements like whey protein or vitamin E. Only two studies [36,38] showed positive outcomes of using fatty acid alone. Six studies showed no differences between the experiment and control groups in terms of muscle strength and mass at the endpoint [25,31,32,34,35,37]. Two studies observed different outcomes in gender, as women had better muscle strength and physical performance after taking the supplement [30,32]. 

#### 3.1.3. Key Findings

Jun et al. [38] found that with the intake of PUFA for 6 months thigh muscle volume increased by 10.3 ± 2.2% (*p* < 0.001), hand-grip strength increased by 10.3 ± 3.3% (*p* < 0.01), and 1-RM muscle strength increased by 9.1 ± 4.1% (*p* < 0.05), while intermuscular fat content decreased by 11.5 ± 5.2% (*p* < 0.05). Gordon et al. [36] also found an increase in thigh muscle volume by 3.6% (*p* < 0.05) compared to the baseline, an increase in 2.3 kg of handgrip strength, and a 4.0% (*p* < 0.05) increase in one rep max (1-RM) muscle strength after 6 months of intervention. No significant outcomes were observed at 3 months. This showed that there was a positive effect of fatty acids over time. Mariasole et al. [32] showed different results based on gender differences, and they found that female participants had an increase in maximal isometric torque (by 18.5% ± 7.2%, *p* < 0.05) and muscle quality (by 18.2% ± 0.5%, *p* < 0.05) after an 18-week intervention, whilst males had an increase in muscle quality by 5.6% ± 5.1% (*p* < 0.05). 

Sebastiaan et al. [33] showed that intake of omega-3 fatty acids for 6 months increased muscle strength over time with leg press 1-RM rising by 30.4% (from 173.6 ± 17.6 to 226.4 ± 21.6 kg, *p* < 0.05); physical performance with 5-repetition chair sit-to-stand test (decreased by 13.7%, *p* < 0.05), 30-s chair sit-to-stand test (increased by 9.7%, *p* < 0.05) and timed up-and-go test (reduced by 6.3%, *p* < 0.05) were also improved. Claire et al. [29] obtained similar results when people took omega-3 for 12 weeks. They found that the anabolic response to amino acid and insulin infusion approximately doubled. Muscle knee extension (test for muscle strength) increased by 2.8 kg (*p* = 0.025) from baseline, and gait speed increased significantly by 8% (*p* = 0.032) from baseline. Calf and thigh muscle thickness increased by 3% to 5% (*p* = 0.018).

These positive outcomes [23,24,29,30,33,35,36,38] reveal that the use of PUFA can improve muscle mass, from 3.6% to 10.3%. With the intake of PUFA and omega-3 for 3 months, muscle strength and physical performance including handgrip strength (increased from 2.3 kg to 3 kg), 1-RM (an increase from 4.0% to 30.4%), and muscle knee extension (increased from 2.0 kg to 2.8 kg) improved significantly.

Conversely, for studies that did not demonstrate significant results [25,31,32,34,37], the used doses (400–800 mg Docosahexaenoic acid, DHA; and 112.5–225 mg Eicosapentaenoic acid, EPA) were different from studies with positive results (400 mg–1.50 g DHA and 270 mg to 1.86 g EPA) which could be a reason for the different outcomes. The studies by Gordon [35] and Roma [34] had a similar experimental design. The participants aged over 60 years old with weak muscle strength or low muscle mass were given different doses of omega-3 (EPA: 1.86 g/d and DHA: 1.50 g/d for 8 weeks in Gordon’s study and EPA: 0.66 g/d and DHA: 0.44 g/d for 12 weeks in Roma’s study) without using resistant training as an intervention. The participants in the study by Gordon et al. [35] were found to have improved muscle protein synthesis rate, whilst participants in Roma et al. [34]’s study had no significant difference in muscle state. Although the number of studies is small, it can signify that the improvement in sarcopenia could be dose-dependent. For more information, refer to Table 1.

### 3.2. Pre-Clinical Studies

There were 10 pre-clinical studies, including 3 cell experiments [14,17,40] and 7 animal studies [18,41,42,43,44,45,46]. Refer to Table 2 and Table 3.

#### 3.2.1. Cell Experiments

One study [14] used the human skeletal muscle myoblast cell line HSMM-1 and found that LCFAs are involved in the pathogenesis of sarcopenia as they activate apoptotic signaling, thereby affecting the myoblast. Two studies [17,40] used mouse skeletal myoblasts cell line C2C12 with *Pistacia lentiscus* L. Seed Oil (PLSO) [40] and eicosapentaenoic acid (EPA) [17] as interventions. Only one study [17] cultured muscle samples from human quadriceps muscle tissue. RT-qPCR, Western blot, and flow cytometry were commonly used in these studies. One study [40] focused on mitochondrial function and showed that PLSO had a positive effect on the prevention of cell death. 

#### 3.2.2. Animal Studies

C57BL/6 mice were used in three studies [41,42,44]. One study [43] used the senescence-accelerated SAMP8 mouse (*n* = 51), two studies [18,46] used Wistar rats, and one used [44] Sprague Dawley rats. Five studies [18,41,43,45,46] had young mice or rats aged from 25 weeks to 8 months as positive control groups, and in the experimental group, elderly mice or rats had an age range from 15 months to 24 months. EPA [41,44], DHA [41], ClA [42], dietary fish oil [45], and olive leaf extracts [18,46] were used as interventions in the animal studies, and all fatty acids showed a positive effect on old mice or rats. One study [43] used trans-fatty acids (from partially hydrogenated vegetable oil, mainly trans-18:1) and found no significant effects on diet or interactions in mice. Two studies [41,43] used the quadriceps muscle for tests, and three studies [18,44,46] used the gastrocnemius muscle instead. Only two studies [42,43] assessed muscle mass and strength. Grip strength was commonly used in measuring muscle physical performance in the studies [43,44,45]. Most studies [18,41,44,45,46] used qPCR to assess gene expression and histological analysis. One study [44] tested the muscle fiber type transition, and two studies [41,42] focused on mitochondrial function.

#### 3.2.3. Key Findings

In cell experiments, EPA showed the ability to prevent cell death and organelle dysfunction, attenuate oxidative stress, and restore differentiation separately. Many cytokines were tested [14,17], including IGF-II, Id3, CASP3, CASP7, CASP9, and BAX, which reveal that fatty acids may impact cell apoptotic and differentiation. The anti-inflammation [17] and antioxidant defense potential [39] were studied separately; they are two of the most studied causes (inflammatory environmental milieu and reactive oxygen species) of sarcopenia. Amarjit et al. [17] also aggravated the symptoms by adding palmitate to cause lipotoxic, induce high levels of cell death, and block myotube formation to show that EPA could partially rescue cell differentiation with enhanced myotube formation being associated with increased MyoD, myogenin, IGF-II, and IGFBP-5 expression (*p* < 0.05).

Amongst these animal studies, EPA [41,44] was proven to increase grip strength, insulin sensitivity, mitochondrial function, and protein quality. The fast-to-slow fiber-type transition in skeletal muscle was inhibited by EPA supplementation. CLA [42] shows an advantage in enhancing mitochondrial adenosine triphosphate (ATP) production and elevating muscle antioxidant enzymes. Olive leaf extracts [18,46] can improve lipid profile, prevent muscle loss and decrease the inflammatory state, but cannot prevent aging-induced alterations. For more details, refer to Table 2 and Table 3.

## 4. Discussion

In older individuals, age-related changes in the body, including sarcopenia, significantly affected the quality of life, clinical outcomes [47], and decreased life expectancy. Fatty acid, a supplement that has been used widely in recent times, plays a vital role in muscle adaptations. Our review systematically assessed 23 studies regarding the relationship between fatty acids and sarcopenia. Amongst the pre-clinical studies, the C2C12 cell line, which is a mouse myoblast cell line, was mainly used to verify the effect of fatty acids. As for animal studies, C57BL/6J mice were mostly used, as well as the senescence-accelerated SAMP8 mouse, which is a promising model for sarcopenia due to its syndrome-like degenerative skeletal muscle atrophy [48]. 

According to previous studies on fatty acids and muscle interaction, fatty acids modulate several aspects of muscle protein adaptations and muscle lipid metabolism. For example, fatty acid oxidation in L6 myotubes increased by 30% after butyrate administration (0.5 mM) [49] and the insulin-independent glucose uptake in C2C12 myotubes was increased after taking propionate [50]. These results show that fatty acids could regulate the transcription of target genes encoding proteins involved in muscle cell lipid metabolism. In vivo studies show that skeletal muscle mass will be influenced after taking short-chain fatty acids (SCFAs) with improved insulin sensitivity [51] and alterations in the antioxidant enzyme activity [52]. Figure 2 shows the effect of fatty acids on metabolic signaling in muscle. The prevailing theory is that fatty acids influence skeletal muscle glucose and lipid metabolism via increased phosphorylation of AMPK and PGC1α which will increase fatty acid oxidation, glucose uptake, mitochondrial content, and oxidative capacity.

Studies reported in this review also showed that fatty acids could increase muscle mass and physical performance by improving mitochondrial function and lipid profile, decreasing the inflammatory state, and even benefiting the restoring differentiation and impacting cell death. These results align with previous findings that fatty acids, including SCFAs, have several positive effects on skeletal muscle mass [51,52], muscle lipid metabolism [53], and anti-inflammatory activities [54] that benefit atrophic skeletal muscle conditions [55]. EPA and DHA have an essential role in reducing inflammation [56,57] and promoting its resolution [58] in the human body. The pre-clinical studies [17,41] in this review also showed that fatty acids could impact the cell death phenotype observed in lipotoxic conditions and improve mitochondrial function. These pre-clinical studies prove that fatty acid is a promising supplement for preventing muscle loss and improving physical performance. For details, refer to Figure 2.

Although pre-clinical studies have shown positive results regarding the outcomes of fatty acids (EPA, DHA, CLA, etc.), strong evidence or clear conclusions were seldom presented in clinical studies. Only two clinical studies [36,38] show positive outcomes using fatty acids alone. However, few studies achieved the expected result, even though in these very studies an RCT [25] that lasted for 36 months failed to have a positive effect. This may mean that the relationship between omega-3 and muscle physical performance is still unclear. Comparing those studies with different results, it can be observed that studies with over 1 g/d of EPA could have positive findings on muscle mass and muscle strength, whilst all studies with less than 1 g/d of EPA had negative findings. To achieve a unified result, the doses of the supplements, duration of interventions, methodologies for assessing body composition, and characteristics of the study populations should be homogenous.

The result from human subjects [33] reveals that the effect of omega-3 fatty acids on muscle may not work through the metabolism of sugar levels. Two studies [30,33] collected blood samples and tested the decrease in IL-6 and TNF-a, which shows that fatty acid will suppress the inflammation cascade in patients. Omega-3 fatty acids results show that despite no changes in muscle volume, resident exercise-induced gains in isometric strength can be further enhanced by omega-3. However, omega-3 did not improve catabolic or inflammatory adaptations. Irrespective of muscle volume, gains in strength (primary criterion for sarcopenia) might be explained by changes in muscle quality due to muscle inflammatory or catabolic signaling [33]. Long-chain n-3 PUFA supplementation augments muscle function, and muscle quality increases in older women but not in older men after exercise training [32]. These findings underline the presence of a gender-specific response, which are to be considered when approaching obesity and related comorbidities.

The strength of this review is that it combines both the pre-clinical and clinical studies on fatty acid and sarcopenia with a reproducible search approach. We explained the potential mechanisms through which fatty acid benefits muscle, substantiating the claims with evidence. A limitation of this review is the lack of quantitative data to perform a meta-analysis due to the heterogeneity of the studies. Some studies use compound nourishment, such as Muscle 5 [23] that contains Vitamin E or Vitamin D [37], which have a positive effect on sarcopenia and thus make it hard to focus on the fatty acid alone. Finally, only one study used the EWGSOP standards; thus, there is a need to include studies with other diagnostic criteria that meet the definitions of sarcopenia definitions for experimental subjects.

## 5. Conclusions

Our systematic review shows that fatty acids will benefit muscles in animals by improving the lipid profile, and enhancing the anti-inflammatory and mitochondrial function. However, more well-designed studies are required to evaluate the dosage and duration of using fatty acids in humans. From this systematic review, it can be stated that intake of omega-3 fatty acids for more than 6 months may be a potential choice for elderlies to prevent or improve sarcopenia as it has positive effects on muscles.

## Figures and Tables

**Figure 1 nutrients-15-03613-f001:**
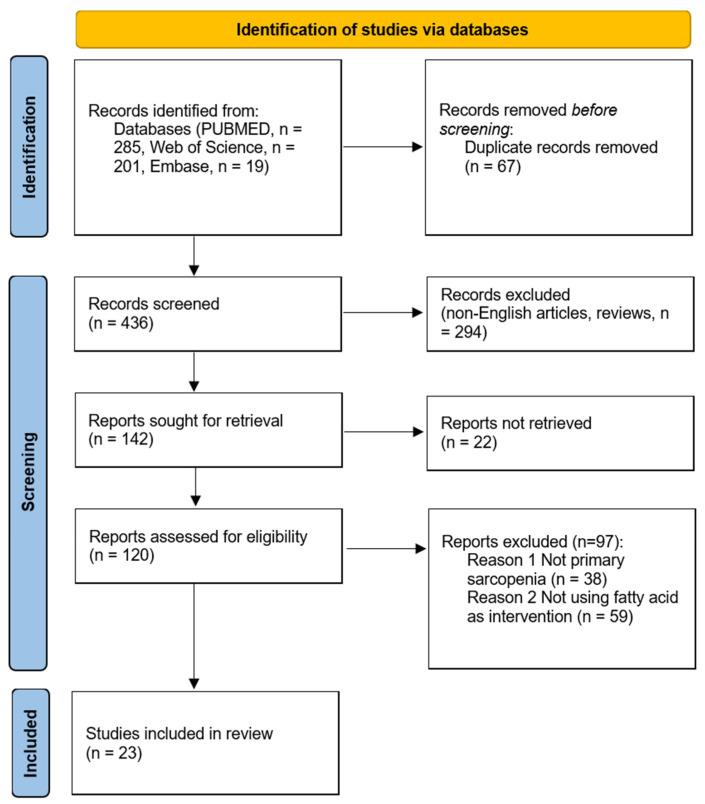
Study search and selection process.

**Figure 2 nutrients-15-03613-f002:**
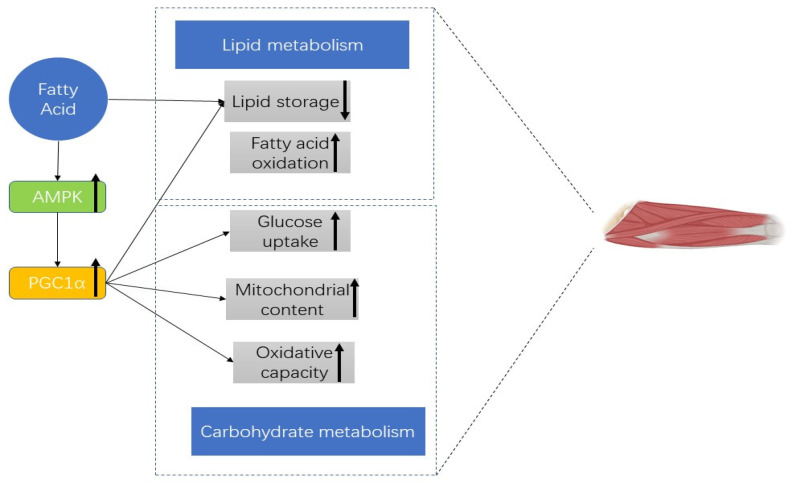
The effects of fatty acids on metabolic signaling pathways in skeletal muscle. (↑: Up regulated. ↓: Down regulated.)

**Table 1 nutrients-15-03613-t001:** Characteristics and key findings of the clinical studies.

Number	Study	Clinical Focus	Experimental Design	Intervention	Duration	Variables Measured	Muscle Measurement	Physical Performance Measurement	Resistance Exercise Training	Blood Sampling/Muscle Biopsy	Results
1	Roma et al. (2015) [34]	53 subjects; 17 men and 34 women (mean age: 74.6 ± 8.0 years)	RCT	n-3 PUFA+ Vit E Vit E (Con)	12 weeks	* MS * MM * MP	bioimpedance analysis	means of the timed up-and-go test (TUG), 4-meter walking test, and handgrip strength	NA	NA	* MS, * MM, * MP: PUFA = Con
2	Mariasole et al. (2017) [32]	50 men and women [men: *n* = 27, age: 70.6 ± 4.5 y; women: *n* = 23, age: 70.7 ± 3.3 y,	RCT	n–3 PUFA placebo (Con), all with lower-limb resistance exercise training	18 weeks	* MP * MS plasma triglyceride concentrations, glucose, insulin, or inflammatory markers	MRI am dmuscle anatomic cross-sectional area (ACSA)	short-performance physical battery (SPPB) knee-extensor muscles	both groups twice a week for 18 weeks	Insulin, * IL-6 and * TNF-a	Decrease in triglyceride In men: * MS, * MP: PUFA = Con In women: MS, MP: PUFA > Con
3	Emelie et al. (2019) [24]	63 healthy recreationally active older women (65–70 years)	RCT	resistance training and healthy diet rich in n-3 PUFAs (RT-HD), resistance training only (RT) and controls (CON)	24 weeks	inflammatory biomarkers MS	NA	NA	twice a a week for 24 weeks	* IL-1β and * mTOR	* MS: RT-HD > RT = Con RT-HD: Decrease in IL-1β and upregulate in mTOR
4	Stephen et al. (2018) [31]	elderly men (*n* = 23); age: 71.4 ± 6.2	RCT	resistance training and omega-3 (Exp) and placebo (Con)	12 weeks	* MM, * MP, blood samples: (* IL-6 and * TNF-α	dual-energy X-ray absorptiometry (DXA)	6 min walk test, chest press, and leg press	3 days per week for approximately 1 h in duration each day with approximately 48 h of rest between resistance training sessions for 12 weeks	* IL-6 ranged from 8.07% to 8.41% and for * TNF-α it ranged from 5.62% to 8.69%.	* MM, * MP: Exp = Con * Pre < * Post * IL-6 and * TNF-α: * Exp = * Con
5	Sebastiaan et al. (2021) [33]	elderly women (*n* = 23)	RCT	resistance exercise and omega-3 (Exp), and corn oil (Con)	12 weeks	* MS * MM * MP inflammatory (p65NF-κB) and catabolic (* FOXO1 and * LC3b) markers	computed tomography scan	5-repetition chair sit-to-stand test (5STS), 30-s chair sit-to-stand test (30STS), timed up-and-go test (TUG), maximal gait speed test (MGS), handgrip strength test (HGS), and leg press.	NA	insulin (* IL-6 blood glucose levels	* MP, * MM, * MS: Exp > Con Decrease in p65NF-κB, * FOXO1, * LC3b
6	Stephan et al. (2020) [37]	13 men and 19 women (age: 60–85)	double-blind, RCT	(1) vitamin D-3 per day. (2) CLA per day. (3) both Vit D and CLA. (4) placebo:	8 weeks	* MS basal myofibrillar protein synthesis	DXA	handgrip strength	NA	myofibrillar protein synthesis, plasma glucose, amino acid, and insulin concentrations	No differences between groups
7	Yves et al. (2019) [25]	Elder (*n* = 1680, men: 1091, women: 589) age:(75.34 ± 4.42 years)	Multi-center, RCT	(1) ω3-PUFA; (2) ω3-PUFA plus the multidomain intervention. (3) the multidomain intervention plus placebo; d) placebo.	36 months	* MS * MP	NA	repeated chair stand test, handgrip strength, walking speed, and balance tests	NA	NA	No differences between groups
8	Jun et al. (2016) [38]	20 healthy participants; 60–85-year-old	double-blind RCT	PUFA corn oil (Con)	6 months	* MS * MM Gene expression	magnetic resonance imaging	handgrip strength 1-RM muscle strength, leg press, chest press, knee extension, and knee flexion	NA	Micro Array	* MS, * MM: PUFA > Con increased the expression of * UCP3 and * UQCRC1
9	Claire. et al. (2020) [29]	41 participants (33 females and 8 males, age: 60–90)	single-center, parallel, double-blind RCT	electrical muscle stimulation procedures + (1) placebo capsules (CHO), (2) whey protein isolate capsules (WPIs), (3) WPI + omega-4 + rutin, and curcumin (WPI + BIO)	1.5 g/d, for 13 weeks	* MP * MM * MS	ultrasonography and MRI	isometric knee extensions, gait speed with three minutes of rest	NA	NA	* MS + *MM: Pre = Post * MP: WPI + BIO > WPI = CHO
10	Gordon. et al. (2015) [36]	44 participants (29 females, 15 males; 60–85-year-old	double-blind RCT	Corn oil (CON) n–3 PUFA group	four 1 g pills/d, for 6 months	* MM * MP	DXA, MRI	handgrip strength, 1-repetition maximum (1-RM) muscle strength, and average isokinetic muscle power	NA	Red blood cell lipids	* MM: Pre < Post * MP: Pre < Post * Body state: Pre = Post
11	Gordon. et al. (2011) [35]	16 older adults (10 men and 6 women, 65 y of age)	RCT	omega-3 fatty acids (Exp) and corn oil (Con)	8 weeks	MM plasma muscle protein fractional synthesis rate (FSR)	DXA	NA	NA	muscle protein synthesis, *Akt, mTOR, and p70s6k, muscle phospholipid fatty acid, and concentrations of phenylalanine and leucine	*MM: Pre = Post FSR: Exp = Con Increased the anabolic response to amino acid Increase in p70s6k and * mTOR
12	Stephen. et al. (2009) [30]	51 elders (28 males and 23 females; age: 65.4 ± 0.8 years)	double-blind, RCT	alpha-linolenic acid (ALA) and placebo (Con)	12 weeks	MS MP cytokines: IL-6, TNF-α	DXA	1-repetition maximum (1RM) chest and leg press strength, knee extensors, and flexors	3 days per week, with at least 1 day of rest between training days, for 12 weeks.	* IL-6 and * TNF-a	* MS: Male in ALA: Pre < Post * MP: ALA = Con IL-6: Male: Pre > Post Female: Pre = Post * TNF-α: Male: Pre > Post Female: Pre < Post
13	Mats et al. (2020) [23]	sedentary men (*n* = 32)	double-blind, RCT	resistance exercise and omega-3 (Exp) and placebo (Con)	12 weeks	* MM * MS * MP	DXA	1-RM grip strength, gait speed, maximal handgrip, isometric knee extension, and SPPB	home-based resistance exercise	NA	* MM, * MS, * MP: Exp > Con; Pre < Post

* List of abbreviations (Appendix A).

**Table 2 nutrients-15-03613-t002:** Characteristics and key findings of the cellular studies.

No.	Study	Cell Line	Intervention	Muscle Samples	Method	Muscle Measurement	Physical Performance	Measurement of Cytokines	Key Findings
1	Amarjit et al. (2017) [17]	C2C12	EPA and a-3 polyunsaturated fatty acid	NA	Interactions of FFAs with TNF-a and IGF-I, qRT-PCR, flow cytometry, and creatine kinase assay	NA	NA	* IGF-II, * Id3, * IGFBP-5, RP-IIb (polr2b), and Myogenin	EPA had little impact on the cell death phenotype observed in lipotoxic conditions but did show benefits in restoring differentiation under lipo-toxic plus cytotoxic conditions.
2	Chen et al. (2020) [14]	muscle myoblast cell line HSMM-1	different concentrations (0, 0.1, 0.3, and 0.6 mM) of PDA	24 paired quadriceps muscle tissue	microarray, RT-qPCR, transfection, immunoprecipitation, chromatin immunoprecipitation (ChIP) assay, and luciferase assay	NA	NA	* FOXM1, PUMA, * BAX, * BAK1, * CASP3, * CASP7, * CASP9, * BCL2, FAS, * NCOR1, * AKT1, and MYC	PDA induced the expression of * FOXM1 and pro-apoptotic genes in vitro, involved in the pathogenesis of sarcopenia by activating apoptotic signaling.
3	Imen et al. (2021) [40]	C2C12 myoblasts	*Pistacia lentiscus* L. seed oil	NA	MTT Assay, fluorescein diacetate assay, plasma membrane permeability, oxidative stress, and mitochondrial function	NA	NA	NA	Prevention of cell death and organelle dysfunction, and attenuation of oxidative stress.

* List of abbreviations (Appendix A); Id3, inhibitor of DNA binding 3; BAK1, BCL2 Antagonist/Killer 1.

**Table 3 nutrients-15-03613-t003:** Characteristics and key findings of the animal studies.

No.	Study	Animal	Intervention	Muscle Samples	Method	Muscle Measurement	Physical Performance	Measurement of Cytokines	Key Findings
1	Matthew et al. (2015) [41]	36 adult (6 months) and 36 old (24 months) C57BL6 mice	chow enriched with EPA or DHA (3.4% kcals) for 10 weeks	quadriceps muscle	mitochondrial energetics, protein fractional synthesis rates, RNA sequencing, and mass spectrometry-based proteomics	NA	NA	NA	EPA can improve mitochondrial function and protein quality but does not restore age-related reductions in mitochondrial protein abundance.
2	Rahman et al. (2009) [42]	80 eleven months old female C57BL/6 (B6) mice	c9t11-CLA or t10c12-CLA for 6 months	whole hind-limb skeletal muscle	mitochondrial function and serum malondialdehyde	dual-energy X-ray absorptiometry (DXA)	NA	* TNF-α and * IL-6	Higher muscle mass, enhanced mitochondrial ATP production, and elevated muscle antioxidant enzymes
3	Jesse et al. (2013) [43]	51 SAMP8 (young, 25 weeks; old, 60 weeks)	trans-fatty acids started from 3 weeks of age	quadriceps muscle	collagen and intramuscular Triacylglycerol content	Echo, MRI system	grip strength and VO2peak	TNF-a	No significant diet effects or interactions.
4	Hiroki et al. (2021) [44]	12 71-week-old C57BL/6J	EPA-enriched diet for 12 weeks	lateral gastrocnemius and plantaris muscle	measurement of triglyceride (TG) levels, RNA sequencing, and histological analysis	NA	grip strength, and treadmill exhaustion test	* MYH7, * MYH2, and * MYH4	Increased grip strength, higher insulin sensitivity, and partially inhibited fast-to-slow fiber-type transition.
5	David et al. (2020) [45]	14 adult (8 months) and 12 aged (22 months) male, Sprague Dawley rats	dietary fish oil for 8 weeks	not mentioned	contractile properties, proteomic analysis, and immunoblotting	NA	grip strength	24 proteins	Increase muscle contractile force, no changes in muscle mass, and no significant associations between contractile parameters and protein abundances.
6	Daniel et al. (2021) [18]	Three months old (Young; *n* = 11) and twenty-four months old (Old; *n* = 17) male Wistar rats	olive leaf extracts and a mixture of algae oil for 21 days	gastrocnemius and soleus muscles	vascular reactivity, serum parameters, protein quantification, qRT-PCR of micro-RNAs, and immunohistochemistry	NA	NA	*Akt, p-Akt, * GSK3β, p-GSK3β	Improved the lipid profile, increased HOMA-IR, and decreased the serum levels of miRNAs 21 and 146a, preventing muscle loss.
7	Daniel et al. (2020) [46]	Young (3 months, *n* = 11) and old (24 months, *n* = 8) male Wistar rats	algae oil and extra virgin olive oil for 21 days	left gastrocnemius muscles	protein quantification and RT-qPCR	NA	NA	* IGF-I, * Akt, LC3b, * HDAC-4	Decreased the inflammatory state, did not prevent aging- induced alterations, and decreased autophagy activity.

* List of abbreviations (Appendix A).

## Data Availability

The results are presented in the paper. For more information, please contact the corresponding author.

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
