# Peer review of "Potential of Fatty Acids in Treating Sarcopenia: A Systematic Review"

_nutrients, 2023, doi:10.3390/nu15163613_

Round 1
Reviewer 1 Report
In this systematic review, Tao HUANG and colleagues the effects of fatty acid supplementation in preventing and treating sarcopenia. They studied a total of 14 clinical studies and 11 pre‐clinical (including cell and animal studies). The overview of these scientific papers helped to elucidate the effects of fatty acids in improving sarcopenia in preclinical experiments. However, current clinical studies showed a controversial results for the role of fatty acids treatment on muscle, and the mechanisms still need to be further studied. The review is interesting, easy to understand and explores an important aspect concerning the loss of muscle mass associated with ageing.
However, I have only minor comments to raise with the authors before considering the review suitable for publication in Nutrients.
Please find below the minor comments:
1- In the introduction section the authors report that long-chain fatty acids (LCFA) are involved in the pathogenesis of sarcopenia by activating apoptotic signalling, while polyunsaturated fatty acids (PUFA) are related to muscle hypertrophy.
I would advise the authors to add some more information on the use of these fatty acids in correlation with sarcopenia in this section.
2- Line 53: Reference 17 does not correspond to the cited article.
3- Line 107-108: Write the acronyms TNF-a and IL-6 in full, they first appear in the text.
4- Line 108-110: Rewrite the concept better, it seems to be an unrelated sentence.
5- Line 189-190: Which type of trans fatty acid is referred to in the text.
6- I find it difficult for the reader to read the text and associate it with the figures in a separate section, I would insert the figures with their captions at the point where they are mentioned.
7- I advise the authors to divide the table between cellular and animal experiments. I would also remind the authors to clarify the acronyms in the caption.
Reviewer 2 Report
Dear Authors,
In my opinion, the topic is interesting; however, I have some doubts regarding the methodological implant of this study, and some critical issues should be addressed.
Major revisions:
INTRODUCTION: This section is too concise and should be improved. In particular, the authors should emphasize and give more importance to the dietary intervention aspect of supplementation.
In relation to this, you should cite the following paper:
- Kunz HE, Michie KL, Gries KJ, Zhang X, Ryan ZC, Lanza IR. A Randomized Trial of the Effects of Dietary n3-PUFAs on Skeletal Muscle Function and Acute Exercise Response in Healthy Older Adults. Nutrients. 2022 Aug 27;14(17):3537. doi: 10.3390/nu14173537.
- Lippi L, Uberti F, Folli A, Turco A, Curci C, d'Abrosca F, de Sire A, Invernizzi M. Impact of nutraceuticals and dietary supplements on mitochondria modifications in healthy aging: a systematic review of randomized controlled trials. Aging Clin Exp Res. 2022;34(11):2659-2674. doi: 10.1007/s40520-022-02203-y.
MATERIALS AND METHODS: MATERIALS AND METHODS: The search strategy should be reported for each database to fulfill the AMSTAR criteria. A Supplementary Table should be provided in the Supplementary Materials.
MATERIALS AND METHODS. The Eligibility Criteria should be clearly reported. The authors should better characterize the inclusion and exclusion criteria. I suggest following the PICO model.
MATERIALS AND METHODS: The outcome measures considered by the authors should be reported and characterized as primary and secondary outcome measures.
MATERIALS AND METHODS: In order to improve the quality of the systematic review, adding a Quality assessment and Risk of Bias Assessment should be considered by the Authors. For performing the quality assessment, I suggest using the Jadad Quality Assessment scale based on the type of studies included. The Risk of bias assessment should be performed using Version 2 of the Cochrane risk-of-bias tool for randomized trials (RoB 2).
RESULTS: A Supplementary Table with excluded studies after full-text examination should be provided. In this table, the authors should report the study ID and the reason for exclusion.
RESULTS: Significant results reported in this section should be supported by their p-values.
Minor revisions:
FIGURES: The PRISMA flow-chart should be updated to the latest version (Page MJ, McKenzie JE, Bossuyt PM, Boutron I, Hoffmann TC, Mulrow CD, et al. The PRISMA 2020 statement: an updated guideline for reporting systematic reviews. BMJ 2021;372:n71. doi: 10.1136/bmj.n71).
TABLES: A list of abbreviations should be provided for tables.
Round 2
Reviewer 2 Report
Dear Authors,
in my opinion, the manuscript is interesting, and the results are intriguing.
You have significantly improved the paper during the revision process.
Therefore, in my opinion, the paper is now suitable for publication in this Journal.
Best regards